# Phytochemical Compounds and Anticancer Activity of *Cladanthus mixtus* Extracts from Northern Morocco

**DOI:** 10.3390/cancers15010152

**Published:** 2022-12-27

**Authors:** Amina El Mihyaoui, Saoulajan Charfi, El Hadi Erbiai, Mariana Pereira, Diana Duarte, Nuno Vale, María Emilia Candela Castillo, Alain Badoc, Ahmed Lamarti, Joaquim C. G. Esteves da Silva, Marino B. Arnao

**Affiliations:** 1Department of Plant Biology (Plant Physiology), Faculty of Biology, University of Murcia, 30100 Murcia, Spain; 2Laboratory of Plant Biotechnology, Department of Biology, Faculty of Sciences, Abdelmalek Essaadi University, Tetouan 93000, Morocco; 3Centro de Investigação em Química (CIQUP), Instituto de Ciências Moleculares (IMS), Departamento de Geociências, Ambiente e Ordenamento do Território, Faculdade de Ciências, Universidade do Porto, Rua do Campo Alegre s/n, 4169-007 Porto, Portugal; 4Biology and Health Laboratory, Department of Biology, Faculty of Science, Abdelmalek Essaadi University, Tetouan 93000, Morocco; 5Biology, Environment and Sustainable Development, Laboratory, Higher School of Teachers (ENS), Abdelmalek Essaadi University, Tetouan 93000, Morocco; 6OncoPharma Research Group, Center for Health Technology and Services Research (CINTESIS), Rua Doutor Plácido da Costa, 4200-450 Porto, Portugal; 7CINTESIS@RISE, Faculty of Medicine, University of Porto, Alameda Professor Hernâni Monteiro, 4200-319 Porto, Portugal; 8Department of Community Medicine, Information and Health Decision Sciences (MEDCIDS), Faculty of Medicine, University of Porto, Rua Doutor Plácido da Costa, 4200-450 Porto, Portugal; 9Laboratoire MIB (Molécules D’Intérêt Biologique), ISVV (Institut des Sciences de la Vigne et du Vin), UMR OENO, Université de Bordeaux, 33140 Villenave-d’Ornon, France

**Keywords:** anticancer activity, *Cladanthus mixtus*, roots, stems, leaves, flowers, MTT assay, PC-3 cells, MCF-7 cells, MRC-5 cells

## Abstract

**Simple Summary:**

Cancer is a dramatic illness that ranks among the most pressing health concerns facing humanity and necessitates a proactive approach to treatment. Phytochemicals are regarded as interesting molecules for the development of anticancer drugs due to their pleiotropic effects on broad targets. The actions of phytochemicals are focused on tumor cells alone, with no negative effects on normal cells. The intricate process of carcinogenesis involves numerous signaling cascades. This study examined the effects of four organ extracts from a Mediterranean endemic plant, *Cladanthus mixtus,* on human tumor cells. To the best of our knowledge, this is the first study on this plant from northern Morocco showing relevant antiproliferative activity against two cancer cell lines and one non-tumoral cell line.

**Abstract:**

Many of the chemotherapeutic drugs for the treatment of cancer are molecules identified and isolated from plants or their synthetic derivatives. This work aimed to identify the bioactive compounds using LC-MS and GC-MS and to evaluate the anticancer activity of the methanolic extracts of roots, stems, leaves, and flowers from *Cladanthus mixtus*. The anticancer activity was evaluated in vitro against two cancer cell lines: human breast carcinoma (MCF-7) and human prostate carcinoma (PC-3), using the MTT assay and microscopic observation. A human normal lung fibroblast (MRC-5) was included to determine the extract’s safety for non-tumoral cells. The chemical composition results by LC-MS analysis revealed the presence of 24 phenolic compounds. Furthermore, GC-MS analysis allowed the identification of many biomolecules belonging to terpenoids, esters, alcohols, alkanes, fatty acids, organic acids, benzenes, phenols, ketones, carbonyls, amines, sterols, and other groups. The findings suggest that the majority of *C. mixtus* extracts have antiproliferative activity against two cancer cell lines, MCF-7 and PC-3, and one non-tumoral cell line, MRC-5. The activity was dose-dependent, and the highest effect was obtained with leaf extract in the two cancer cell lines. Moreover, these extracts demonstrated an acceptable toxicological profile against normal cells. Overall, *C. mixtus* extracts revealed promising antitumor properties provided by their phytochemical composition.

## 1. Introduction

Cancer is one of the leading causes of death worldwide [1]. According to the World Health Organization (WHO), cancer caused 10 million deaths in 2020 [2]. Female breast cancer, the most common cancer, was estimated at 2.26 million (11.7%) new cases, followed by lung cancer (2.21), colorectal cancer (1.93), prostate cancer (1.41), and stomach cancer (1.09 million) with percentages of 11.4, 10.0, 7.3, and 5.6%, respectively. According to global cancer incidence predictions, there will be 28.4 million new instances of cancer worldwide in 2040, up 47% from 2020 [3].

Cancer arises from the transformation of normal cells into tumor cells, a multi-step process that usually begins with a precancerous lesion, which then becomes a malignant tumor, according to the WHO in 2022. This can be caused by genetic mutations that stimulate uncontrollable cell division, either locally or through metastasis [4]. DNA damage, whether exogenous or endogenous, is the origin of this mutation. Under normal circumstances, damaged cells would initiate a repair procedure that incorporates cell cycle arrest to prevent further damage. Among the most active tumor suppressor genes, P53 works to control cell cycle arrest and, if necessary, triggers apoptosis [5]. This procedure stops the transmission of mutations to cells’ subsequent generations [6]. Nevertheless, when this system is broken, cancer cell mutation happens very quickly, which results in the formation of tumors [7].

A great source for treating a variety of human ailments is medicinal plants. Due to their enormous chemical and biological diversity and the existence of several substances with promising biological activity, they are currently the focus of numerous recent studies [8]. Chemotherapeutic drugs for the treatment of cancer are molecules identified and isolated from plants or their synthetic derivatives [9]. Many scientific studies have suggested that some terpenoids from plants may be able to destroy cancer cells [10,11,12]. Further, phenolic compounds, particularly flavonoids, have long been reported as chemopreventive agents in the treatment of cancer [13,14].

*Cladanthus mixtus* (L.) Chevall. (synonymous with *C. mixtus* (L.) Oberpr. and Vogt., *Ormenis mixta* subsp. *mixta*, *Anthemis mixta* L., *Chamaemelum mixtum* (L.) All., Moroccan chamomile, and simple leaved chamomile) [15], is an Asteraceae Anthemidae Santolininae [16], which is a traditional herbal medicine widespread in Morocco, Algeria, and the northern and eastern parts of the Mediterranean basin. Moroccan chamomile leaves and flowers have been used in traditional Moroccan medicine as an infusion to treat various ailments [17,18]. Moreover, *C. mixtus* is used by naturopaths and herbalists as an analgesic, antiallergic, anti-inflammatory, antispasmodic, carminative, digestive, febrifuge, fungicide, sudorific, vermifuge, stimulant of leukocyte production [19], and antioxidant agent [20]. Generally, there are a few works on the traditional use of this plant around the world. To the best of our knowledge, this is the first study on this plant from northern Morocco (Tangier-Tetouan-Al Hoceima region).

The body of research on the therapeutic effect of *C. mixtus* continues to grow due to its reported interesting biological activities, which appear to be related to its bioactive compounds. The main objective of our study was to investigate the chemical composition of the four *C. mixtus* organs (roots, stems, leaves, and flowers) using LC-MS and GC-MS analysis and to demonstrate that these four *C. mixtus* extracts can induce suppression of cancer cell growth.

## 2. Materials and Methods

### 2.1. Standards and Chemical Reagents

Phenolic standards such as salicylic acid, syringic acid, ferulic acid, gallic acid, *p*-hydroxybenzoic acid, *p*-coumaric acid, protocatechuic acid, vanillic acid, rosmarinic acid, caffeic acid, ellagic acid, chlorogenic acid, catechin, luteolin-7-glucoside, luteolin, apigenin-7-*O*-glucoside, apigenin, quercetin, rutin, kaempferol, isorhamnetin, naringin, methyl paraben, and vanillin were purchased from Sigma-Aldrich (Darmstadt, Germany). Solvents and reactants such as acetonitrile, acetic acid, ethyl acetate methanol, *N*,*O*-bis(trimethylsilyl)trifluoroacetamide (BSTFA) pyridine, and thiazolyl blue tetrazolium bromide (MTT, cat. no. M5655) were obtained from Merck KGaA (Darmstadt, Germany). Dulbecco’s Modified Eagle’s Medium (DMEM), fetal bovine serum (FBS), and penicillin–streptomycin (pen–strep) solution were obtained from Millipore Co. (Merck KGaA, Darmstadt, Germany). Other cell culture reagents were purchased from Gibco (Thermo Fisher Scientific, Inc, Waltham, MA, USA).

### 2.2. Plant Material

*Cladanthus mixtus* (L.) Chevall. plant was collected at full maturity from Beni Hassane, Tanger-Tetouan-Al Hoceima region, northern Morocco (N 35°21′20.865″, W 5°22′12.677″) in May 2018 and transported to the laboratory. Plant identification was performed by Prof. Lamarti in Tetouan (Morocco). Botanical characteristics can be consulted on the taxon page of World Floral Online (WFO, 2022): *Cladanthus mixtus* (L.) Chevall. Published on the Internet; http://www.worldfloraonline.org/taxon/wfo-0000122557, accessed on: 23 November 2022. Human cell lines were provided by the American Type Culture Collection com-pany (ATCC; Manassas, VA, USA), including MRC-5 human normal lung fibroblast, MCF-7 human mammary carcinoma, and PC-3 human prostate carcinoma cell lines. After separating the organs (roots, stems, leaves, and flowers) of fresh plants, the ma-terial was dried in an oven until stabilization of dry weight at 50 °C in order to pre-serve the integrity of its natural composition as much as possible. Then, it was ground in a Microtron-MB550 (Kinematica AG, Germany) at 8000 rpm. The powder obtained was composed of particles with a diameter of around 0.2 mm and stored in the dark at room temperature.

### 2.3. Analysis of Phenolic Compounds by HPLC-MS

#### 2.3.1. Extraction of Phenolic Compounds from *Cladanthus mixtus*

A quantity of 1 g of powder from each organ was extracted with 20 mL of methanol:water (80:20, *v/v*) at −20 °C for 2 h in static condition. After sonication for 15 min and centrifugation at 4000× *g* for 10 min, the extract was filtered through Whatman N^o^ 4 paper. The residue was then re-extracted under the same conditions once more before evaporating it. Then, the extracts were combined and evaporated at 40 °C under reduced pressure using a rotavapor (Rotavapor^®^ R-210, BÜCHI, Flawil, Switzerland) to remove the methanol. The aqueous phase was subjected to another liquid–liquid extraction with diethyl ether (2 × 20 mL) and ethyl acetate (2 × 20 mL). Sodium sulfate anhydrous was added to the organic phase during decantation, and the extracts were filtered through Whatman N^o^ 4 paper before being evaporated by a rotary evaporator at 40 °C to dryness. For HPLC-MS analysis, a quantity of 10 mg of each dried extract was dissolved in 1 mL of methanol:water (80:20, *v/v*) and then filtered with a 0.22 µm disposable LC filter disk [21].

#### 2.3.2. Analysis by HPLC-MS

The analysis of phenolic components was performed using a procedure previously described by Erbiai et al. [21] utilizing similar settings and HPLC equipment. Briefly, the phenolic extract was analyzed using a high-performance liquid chromatograph Finnigan Surveyor Plus equipped with a PDA Plus detector and a Plus autosampler (Thermo Scientific, Waltham, MA, USA). Chromatographic separation was accomplished using an Acclaim™ 120 (Thermo Scientific, Waltham, MA, USA) reverse phase C18 column (3 µm 150 *×* 4.6 mm) thermostatted at 35 °C, and peaks were detected at 280 nm as the preferred wavelength. The mobile phase was 1% acetic acid in water (A) and 100% acetonitrile (B). The elution gradient established was from 10 to 15% B over 5 min, from 15 to 25% B over 5 min, from 25 to 35% B over 10 min, from 35 to 50% B over 10 min, isocratic 50% B for 10 min, and re-equilibration of the column, using a flow rate of 0.5 mL/min. LC–MS analysis was performed using an LC quaternary Plus pump coupled to a Finningan LCQ Deca XP MAX mass detector with an electrospray ionization (ESI) source and an ion trap quadrupole (Thermo Scientific, Waltham, MA, USA).

The identification of phenolic compounds was performed according to the UV-Vis spectra. The mass spectra and retention times were used in comparison with commercial standards. The area of the peaks recorded at 280 nm was used for quantification by comparison with calibration curves obtained from the standard of each molecule. The results were expressed in µg per gram of dried weight (µg/g DW). All the chemical analyzes were carried out in the Chemistry Laboratory at the Faculty of Sciences, Porto University (Portugal).

### 2.4. Preparation of Methanolic Extracts

Extraction by maceration of different organs from *C. mixtus* (roots, stems, leaves, and flowers) with methanol was carried out according to the procedure used by Barros et al. [22] and modified by Erbiai et al. [23] with slight modifications. Two grams of powder from each organ were extracted by shaking with 100 mL of methanol at 25 °C at 150 rpm for 24 h and then vacuum filtered through Whatman N^o^ 4 paper. The filtration residue was re-extracted twice again using the same procedure. The methanolic extracts of each organ were combined and then evaporated at 45 °C to dryness using a rotary evaporator under vacuum (Rotavapor^®^ R-210, BÜCHI, Flawil, Switzerland). Then, the dried extracts of roots (CM-R), stems (CM-S), leaves (CM-L), and flowers (CM-F) were weighed and stored at −80 °C until use.

### 2.5. GC-MS Analysis of the Methanolic Extracts from Cladathus mixtus

Gas chromatography (GC) (Trace 1300 gas chromatograph, Thermo Fisher Scientific, Waltham, MA, USA) coupled to mass spectrometry (MS) (ISQ single quadrupole mass spectrometer; Thermo Fisher Scientific) was used to conduct a chemical analysis of the methanol extracts of *C. mixtus* (CM-R, CM-S, CM-L, and CM-F). The GC was equipped with a capillary column DB-5 (Thermo Fisher Scientific, Waltham, MA, USA) (30 µm, 0.25 mm i.d, film thickness 0.25 µm) with a non-polar stationary phase (5% phenyl, 95% dimethylpolysiloxane). The column temperature was programmed from 50 to 350 °C at a rate of 5 °C/min. Helium was used as a carrier gas at a flow rate of 0.75 mL/min [23]. For GC–MS analysis, 10 mg methanolic extracts were dissolved in 1 mL chloroform.

### 2.6. In Vitro Anticancer Studies

#### 2.6.1. Cell Culture

MRC-5 human normal lung fibroblast, MCF-7 human mammary carcinoma, and PC-3 human prostate carcinoma cell lines in passages 8, 44, and 16, respectively, were used for the evaluation of biosafety of the Moroccan chamomile extracts studied in this work. These cell lines were obtained from the American Type Culture Collection (ATCC; Manassas, VA, USA) and maintained in DMEM cell culture medium supplemented with 10% FBS and 1% pen–strep solution at 37 °C and 5% CO_2_. For passaging, confluent cells were trypsinized using a 0.25% trypsin–EDTA solution and subcultured in the same culture media. Prior treatments, 8000 (MCR-5), 10,000 (MCF-7), and 5000 (PC-3) cells/well were seeded in 96-well plates and allowed to adhere for 24 h.

#### 2.6.2. Cell Treatment

The biological effect of four chamomile extracts (CM-R, CM-S, CM-L, and CM-F) was evaluated after 48 h of treatment using 96-well plates. The different cell lines were incubated with each extract in the following concentrations: 50, 100, 200, and 500 µg/mL. Stock solutions were prepared in 100% dimethyl sulfoxide (DMSO) and were diluted to 0.1% DMSO on the day of the experiment in cell culture media. Control cells were treated with 0.1% DMSO (vehicle). Their toxicological effect was evaluated using morphological analysis and MTT assay.

#### 2.6.3. Morphological Analysis

Cellular morphology in monolayer culture was captured after each treatment of 48 h using a Leica DMI 6000B microscope coupled with a Leica DFC350 FX camera (Leica Microsystems, Wetzlar, Germany). Images were treated with the Leica LAS X imaging software (v3.7.4) (Leica Microsystems, Wetzlar, Germany).

#### 2.6.4. MTT Assay

The cytotoxic profile of each extract was evaluated with an MTT assay. Briefly, after each treatment during 48 h, cell media were aspirated, and 100 µL of MTT solution (0.5 mg/mL in PBS) was added to each well. Plates were protected from light and incubated for a period of 3 h at 37 °C. Then, MTT solution was aspirated from each well and replaced with 100 µL/well of DMSO to solubilize the formazan crystals. Absorbance was measured at 570 nm using an automated microplate reader (Tecan Infinite M200, Tecan Group Ltd., Männedorf, Switzerland).

### 2.7. Statistical Analysis

GraphPad Prism 9 (GraphPad Software Inc., San Diego, CA, USA) was used. The results of three independent experiments are represented as the mean ± SEM for antiproliferation activity and ± SD for chemical analysis in the three cell types assayed. Statistical analysis was performed with one-way ANOVA tests by Dunnett’s multiple comparisons between control and treatment groups. Statistical significance was accepted at *p* values < 0.05.

## 3. Results

### 3.1. Phenolic Acids and Flavonoids Characterized in Cladanthus Mixtus by HPLC-MS

We studied the presence of phenolic compounds in *C. mixtus* extracts using 24 standards. Appendix A shows the chromatogram, and Appendix A illustrates the quantification of the phenolic compounds. Results of HPLC-MS analysis of *C. mixtus* extracts showed the presence of 23 phenolic compounds identified in the CM-F and 24 compounds in the CM-L, CM-S, and CM-R extracts (Appendix A). Although showing differences in their phenolic composition, all extracts were characterized by a high content of phenolic acids and flavonoids. As detailed in Table 1, statistically significant differences were found between the content of the identified molecules in the same organ (*p* < 0.05). Regarding phenolic acids, all 13 compounds were identified and quantified in the extracts of *C. mixtus* organs, except for vanillic acid, which was absent in CM-F. Chlorogenic acid was the main phenolic acid determined at high concentrations in CM-F (1987.02 µg/g DW), CM-S (894.49 µg/g DW), and CM-R (561.82 µg/g DW), while ellagic acid was found to be the most abundant phenolic acid in CM-L (1095.01 µg/g DW). However, gallic acid showed the lowest concentration in all organs (ranging from 7.14 to 13.23 µg/g DW). Generally, flowers showed the highest content for all phenolic acids in comparison to the other ones.

Concerning flavonoids, all 11 molecules were characterized in the four organs, with a dominance of glycosides and aglycones (Table 1). Apigenin-7-*O*-glucoside had the highest content in CM-L, CM-S, and CM-R with values of 958.59, 819.60, and 603.02 µg/g DW, respectively, while it was classified as a second major compound following quercetin (1292.01 µg/g DW) in CM-F. In contrast, apigenin was observed to have a higher content in both CM-F (119.14 µg/g DW) and CM-S (29.83 µg/g DW), while quercetin and kaempferol were characterized as the lowest flavonoids in CM-L and CM-R with the values of 23.38 and 39.90 µg/g DW, respectively. Similar to phenolic acids results, flowers showed the highest content for all flavonoids compared to other organs.

### 3.2. Biochemical Constituents of the Different Organs of Cladanthus mixtus by GC-MS

The GC-MS chromatograms of CM-F, CM-L, CM-S, and CM-R (Appendix A) at different retention times revealed the presence of 38, 42, 30, and 36 phytochemicals, respectively (Table 2). These biomolecules can be divided into several groups, including terpenoids, esters, alcohols, alkanes, fatty acids, organic acids, benzenes and their derivatives, phenols, ketones, carbonyls, amines, sterols, and other groups. In general, CM-F was dominated by fatty acids (27.86%), CM-L by terpenoids (46.20%), CM-S by esters (30.11%), and CM-R by alcohols (24.49%) and esters (21.91%).

As presented in Appendix A, GC-MS analysis of *C. mixtus* methanolic extract showed that terpenoids represented 46.2% of CM-L composition, 11.32% of CM-S composition, 10.88% of CM-R composition, and 10.41% of CM-F composition. CM-L contained 10 compounds, dominated by lupeol (14.66%) and phytol (11.39%). Four compounds were detected in both CM-F and CM-S, with β-sitosterol (6.25%) and phytol (6.41%) as the major terpenoid compounds, respectively. From the three biomolecules identified in CM-R, lupeol represented the higher percentage in this organ, with a value of 8.69%.

Regarding esters, as shown in Appendix A, the results presented values ranging from 13.21 to 30.11%, with the order: CM-S ˃ CM-R ˃ CM-F ˃ CM-L. For CM-F, CM-L, and CM-R, the results showed the presence of 5 compounds, while CM-S contained 6 compounds. Palmitic acid β-monoglyceride was detected as the major constituent in all extracts (15.66, 13.08, 7.88, and 6.32% in CM-S, CM-R, CM-F, and CM-L, respectively), followed by stearic acid β-monoglyceride (5.57, 4.00, 3.29, and 3.21% in CM-R, CM-S, CM-F, and CM-L, respectively).

Concerning alcohol composition, the results differed widely between the studied organs (Appendix A). CM-F did not contain alcohols, while the other organs showed specific and unique compounds. Three alcohols were found in CM-R (24.49%), dominated by ethyl iso-allocholate (23.28%). Similarly, CM-L contained 3 alcohols (6.74%), with diallyl methyl carbinol (5.39%) as the main compound. On the other hand, 1-cyclohexanol, 1-[5-hydroxy-4-methyl-2-hexenyl] (3.98%) and 5-azacyclodecanol (1.07%) were the two alcohols detected in CM-S.

In *C. mixtus* extracts, alkanes represented 18.13, 15.14, 3.14, and 2.63% in CM-S, CM-F, CM-R, and CM-L, respectively (Appendix A). Eicosane was the major compound reported in CM-F (13.46%) and CM-L (2.14%), while 1,3,5-trimethyl-2-octadecylcyclohexane was predominant in CM-S (6.17%) and CM-R (2.26%).

For the fatty acids, CM-F showed the presence of 5 compounds, with (*Z*)-18-octadec-9-enolide (12.69%) as a major compound, followed by palmitic acid (9.10%) (Appendix A). The latter was predominated in CM-L with a value of 11.39%. For CM-R and CM-S, (*Z*)-13-docosenamide was the only fatty acid reported, with 8.23 and 6.18%, respectively. Concerning the composition of the organic acids (Appendix A), three of them were found in CM-L, two organic acids in CM-R and CM-F, and only one organic acid in CM-S. Oxalic acid dihydrazide was the only common compound among the four organs, showing the highest percentage in CM-R (3.47%) and CM-F (1.88%).

Furthermore, the GC-MS analysis of *C. mixtus* extracts revealed the presence of benzene and its derivatives, with values ranging from 2.85 to 8.15% (Appendix A). CM-R contained only 1-nitro-3-(propoxymethyl) benzene (8.15%). CM-F and CM-S were formed of two compounds, with loliolide (benzofuran) as the dominant compound. The latter was predominated in CM-L, containing a total of three molecules. Additionally, *C. mixtus* extracts also contained phenols, ranging from 0.41 to 6% (Appendix A). CM-R contained three of them, with vanillin as the dominant compound (4.21%), followed by 2-methoxy-4-vinylphenol (1.28%). The latter was the unique phenol detected in CM-L (0.52%). CM-S and CM-F also contained only one phenol, 6-*O*-acetyl-1-[[4-bromophenyl] sulfonyl]-β-D-glucoside (1.61%) and 4-hydroxy-2-methylacetophenone (0.41%), respectively.

On the other hand, flowers did not contain ketones, while the other organs showed different compositions, as shown in Appendix A: CM-S (4.47%, two compounds), CM-R (1.67%, two compounds), and CM-L (1.25%, one compound). Concerning the carbonyl group, it was represented in CM-F by 5-ethyl-4-methyl-3-heptanone (19.27%) and in CM-L by 4-heptanol and 4-ethyl-2,6-dimethyl (1.34%) (Appendix A). Moreover, the four organs showed a different composition of amines (Appendix A). CM-S was formed by *N*-butylcyclohexylamine (6.90%) as a single and main biomolecule. CM-F and CM-R also contained one molecule, while CM-L showed two amine compounds (2.32%).

The results also showed the presence of pyrrolidines in *C. mixtus* extracts (Appendix A). CM-S and CM-F showed three compounds with a total of 3.04 and 0.97%, respectively. On the other hand, CM-R contained two pyrrolidines (1.29%), while CM-L contained only one (0.22%). Regarding pyrimidines, a single compound was detected only in CM-R, 2(1H)-pyrimidinethionetetrahydro-1,3-dimethyl (2.07%) (Appendix A). Similarly, one steroid was present only in CM-L, 2-methylene-5-α-cholestan-3-β-ol (2.32%).

About the rest of the compounds detected by GC-MS (Appendix A), there was a difference in the constituents in all organs. A variety of groups have been observed, including aldehydes, coumarins, ethers, siloxanes, organosulfur, and organophosphate. In CM-R, six molecules (6.77%) were reported. In CM-S, two molecules formed 4.72% of the extract. In CM-F and CM-L, three molecules were detected.

### 3.3. Cytotoxic Effect In Vitro of Cladanthus mixtus Extracts

We analyzed the antitumor effect of the four methanolic extracts of *C. mixtus* CM-R, CM-S, CM-L, and CM-F against human mammary carcinoma MCF-7 and human prostate carcinoma PC-3 cell lines, to study their efficacy against these types of cancer cells. To assess their biosafety profile, these extracts were also evaluated in a human normal lung fibroblast MRC-5. Cells were treated with DMSO and increasing concentrations of each extract (0–500 µg/mL) to evaluate cell viability after 48 h treatment. Cell viability was assessed with MTT assay, along with changes in cell morphology.

#### 3.3.1. Human Normal Lung Fibroblast (MRC-5)

Based on the MTT results against MRC-5, CM-R and CM-S did not cause a significant reduction in the viability of MRC-5 cells (Figure 1A,B, respectively) in the range of concentrations tested. However, a decrease of about 10% in cell viability was found in cells treated with CM-R for concentrations above 100 μg/mL (Figure 1A) and above 50 μg/mL in cells treated with CM-S (Figure 1B). On the other hand, a more significant decrease was detected by CM-F (*p* < 0.05) (Figure 1D) and CM-L (*p* < 0.001) (Figure 1C) at 500 µg/mL. In addition, microscopic analysis of all extracts at 500 μg/mL showed suppression of proliferation and fewer cell numbers compared to the vehicle (DMSO) (Figure 2), supporting MTT results. Taken together, these results demonstrate that these extracts are safe for normal cells in concentrations until 200 μg/mL.

#### 3.3.2. Human Breast Carcinoma Cell (MCF-7)

The results of the MTT assay for MCF-7 showed that CM-S and CM-F had little activity on cell viability (Figure 3B,D), while CM-R decreased the cell viability at all tested concentrations compared to control, with better results at the highest value (500 μg/mL) (Figure 3A). On the other hand, CM-L at 500 µg/mL displayed a significant antitumor activity (*p* < 0.01), causing a reduction of 50% of MCF-7 cells (Figure 3C). Microscopic analysis indicated that the cells were rounder and in lesser numbers, without the formation of aggregates for CM-L and CM-R (Figure 4). The other extracts did not cause any changes in cancer cell morphology.

#### 3.3.3. Human Prostate Carcinoma Cell (PC-3)

Against PC-3 cells, CM-R showed significant antitumor activity at 200 µg/mL (*p* < 0.001) and 500 µg/mL (*p* < 0.0001) (Figure 5A). CM-S (Figure 5B) and CM-F (Figure 5D) showed significant toxicity at 500 µg/mL (*p* < 0.01), while CM-L revealed a more significant activity at 500 µg/mL (*p* < 0.001) (Figure 5C). Morphological evaluation of PC-3 cells treated with *C. mixtus* extracts showed a decrease in cell number and smaller and rounder cells at 500 µg/mL compared to control cells (vehicle) (Figure 6).

## 4. Discussion

Anticancer drugs destroy cancer cells by stopping their growth or multiplication. However, some chemicals are non-selective and can target a significant portion of healthy cells. For this reason, investigation of cytotoxic substances from natural resources such as plants is necessary [24]. Therefore, the availability of natural products with higher efficacy and fewer side effects is desired. Medicinal herbs have been widely used for the treatment of diseases traditionally for several generations. An interaction between traditional medicine and modern biotechnological tools must be established for the development of new drugs [25]. Bioactive compounds are responsible for the medicinal characteristics of plants and exhibit biological activities [26].

This study aimed to evaluate the anticancer property of methanolic extracts from four organs of the aromatic and medicinal plant *C. mixtus* against two cancer cell lines: MCF-7 (human breast carcinoma) and PC-3 (human prostate carcinoma) and one non-tumoral cell line: MRC-5 (human normal lung fibroblasts), using the MTT assay. Overall, the results of the antitumor activity were very encouraging, showing a significant decrease in cell viability in the cancer cell lines in a concentration-dependent way. In general, the most promising extracts (CM-L and CM-F) are also the most cytotoxic to normal cells. Both extracts work most effectively in cancer cells, in the same concentrations, in which a significant decrease in the survival of MRC-5 cells was obtained. Moreover, cell survival of the MCF-7 line was better than that of the MRC-5 line. Only cells of the PC-3 line showed greater sensitivity to the tested extracts. By comparing the four studied organs of *C. mixtus* on MRC-5 cells (Figure 7A), we can report that they have an acceptable biosafety profile and do not induce a significative reduction in cell viability in concentrations under 500 µg/mL of extracts. Additionally, the best antiproliferative activity on MCF-7 cells was reported when using the highest concentration (500 μg/mL). Leaves (CM-L) showed the most significant effect (*p* < 0.01), followed by root (CM-R) and flower (CM-F) extracts (Figure 7B). Furthermore, comparative cytotoxic evaluation of *C. mixtus* extracts in the PC-3 cell line revealed that leaf and flower extracts provided the most significant anticancer activity (*p* < 0.001 and *p* < 0.01, respectively), with an almost 50% reduction in cell viability at a concentration of 500 µg/mL (Figure 7C). Consequently, leaf extracts showed the highest anticancer activity against the two selected cancer cell lines. This explains the use of leaves in most preparations in traditional Moroccan medicine as phytotherapeutic treatments [17,18]. Moreover, special attention should be paid to the CM-R extract, which was the only one that generated a decrease in cancer cell viability at a concentration of 200 µg/mL while being safe for normal MRC-5 cells.

The antiproliferative activity of medicinal plants is related to their chemical composition. Natural phenolic compounds have shown their ability to modulate cell signaling pathways linked to cell death [27,28]. For this reason, we carried out a qualitative and quantitative determination of the phenolic compounds of *C. mixtus* using an HPLC-MS analysis. The hydroalcoholic extracts (methanol 80%) of the four organs were prepared in order to extract free and conjugated phenolic compounds with hydrophilic and lipophilic characters. The results showed that our extracts contained mainly flavonoids in co-presence with derivatives of cinnamic acid. Twenty-four phenolic compounds were identified (gallic acid, protocatechuic acid, chlorogenic acid, salicylic acid, *p*-hydroxybenzoic acid, caffeic acid, vanillic acid, syringic acid, rosmarinic acid, ellagic acid, ferulic acid, *p*-coumaric acid, apigenin, luteolin, apigenin-7-*O*-glucoside, luteolin-7-*O*-glucoside, quercetin, rutin, naringin, methyl paraben, catechin, vanillin, kaempferol, and isorhamnetin) differentially concentrated in the extracts of *C. mixtus*. The absence of vanillic acid was noted in the flower extract. In this context, it should be emphasized that all analyzed extracts of *C. mixtus* contained an interesting amount of apigenin-7-*O*-glucoside ranging from 603 to 1074 µg/g DW, and the highest value was represented in flowers. Our data are according to the study of Elouaddari et al. [29], which qualitatively demonstrated the presence of gallic acid, catechin, vanillic acid, caffeic acid, and syringic acid in the aerial parts of *C. mixtus*. Haghi et al. [30], applying UPLC (ultra-performance liquid chromatography) coupled to a photodiode array (PDA) for the separation of flavonoids from *Matricaria chamomilla*, another chamomile of the Asteraceae family, demonstrated that the content of apigenin-7-*O*-glucoside in crude extracts was much higher than free apigenin, which is similar to our results. Additionally, Elsemelawy [31] indicated that flowers and roots extracts of chamomile (*M. chamomilla*) were rich in phenolic acids (ellagic acid, catechol, and chlorogenic acid with 1582.81, 1104.49, and 937.48 ppm, respectively) and flavonoids (luteolin *O*-acylhexoside and quercetin with 2801.99 and 1765.01 ppm, respectively). On the other hand, methanolic extract from Iranian chamomile (*M. chamomilla*) presented flavonoids such as luteolin (2.2 mg/g) and apigenin (1.19 mg/g) [32], which were also detected in our chamomile.

Several molecules found in our plants are known for their anticancer effect. Phenolic acids are a subclass of plant phenolic compounds, divided into benzoic acid (*p*-hydroxybenzoic acids including vanillic, protocatechuic, syringic, and gallic acid) and cinnamic acids (including ferulic, *p*-coumaric, caffeic, and sinapic acid), which are associated with potent anticancer activity in various in vitro and in vivo studies [33]. Vanillic acid has free-radical scavenging antioxidant activity, serves as a chemoprotectant, and helps to prevent benzo(a)pyrene-induced lung cancer in Swiss albino mice [34]. In the 7,12-dimethylbenz[a]anthracene (DMBA)-induced hamster buccal pouch carcinogenesis, vanillic acid also showed antioxidant and antilipidic peroxidative properties. It is interesting to note that the administration of vanillic acid restored lipid peroxidation by-product levels and aberrations in antioxidative status, which was observed to have changed after administering DMBA alone [35]. Moreover, protocatechuic acid (a dihydroxybenzoic acid) has several biological activities, such as antibacterial, antiviral, antidiabetic, antiulcer, antifibrotic, analgesic, anti-inflammatory, cardiac, antiaging, hepatoprotective, neurological, nephroprotective, antioxidant, and anticancer effects [36]. In cancer cell lines, protocatechuic acid caused apoptosis and prevented tumor cell growth via several mechanisms [37]. Indeed, in human breast, prostate, liver, lung, and cervix cancer cells, this compound caused apoptosis, suppressing invasion and metastasis through enhancing DNA fragmentation, suppressing interleukins levels (IL-6 and IL-8), increasing caspase-3 and -8 activity, decreasing matrix metalloproteinases (MMP), intercellular adhesion molecule 1 (ICAM-1) level, vascular endothelial growth factor (VEGF) level, and Na^+^-K^+^-ATPase activity [38]. On the other hand, syringic acid exhibited high and dose-dependent antitumor activity against DMBA-induced hamster buccal pouch carcinogenesis [39]. In human lung A549 and colon adenocarcinoma HT29-D4 cells, caffeic acid exhibited an antioxidant effect by inhibiting ROS-SOD production, and it prevented cancer progression and migration by reducing cell attachment to the extracellular matrix, thus decreasing cell adhesion [40,41]. In prostate cancer PC-3 and LNCaP cell lines, ferulic acid inhibited cell proliferation and invasion and induced apoptosis at 300 and 500 µM, respectively [42,43]. Additionally, in the TT human thyroid cancer cell line, ferulic acid exhibited anti-carcinogenesis activity through decreasing the expression of novel gene URG4/URGCP, CCND1, CDK4, CDK6, BCL2, MMP2, and MMP9 and increasing the expression of p53, Poly(ADP-ribose) polymerase (PARP), PUMA, NOXA, BAX, BID, caspase-3 and -9, and TIMP1 genes significantly. This caused the suppression of invasion, migration, and colony formation [44]. Furthermore, *p*-coumaric acid demonstrated remarkable scavenging of free radicals and NF-κB modulatory effects [45]. In addition, in both in vitro and in vivo models of colon cancer, *p*-coumaric acid treatment activated the unfolded protein response-mediated apoptosis and inhibited glucose-related protein 78, which is frequently dysregulated in colon cancer [46].

Moreover, flavonoids have long been reported as chemopreventive agents in the treatment of cancer [13,14,47,48]. Indeed, flavonoids have exhibited anticancer activity [49], along with anti-inflammatory [50], antioxidant [51], and antimicrobial properties [52]. Among flavonoids found in our extracts, apigenin showed high biological and pharmacological activities [53], such as antimutagenic and antiproliferative activities in several cancer cell lines [54]. In addition, quercetin has shown promising anticancer activity against breast cancer in the MCF-7 xenograft model [55]. It also inhibited breast cancer invasion by increasing the expression of pro-apoptotic protein Bax and caspase-3, as well as decreasing oncogenic EGFR in MDA-MB-231 cells and MCF-7. Another in vivo study reported that quercetin significantly reduced tumor diameter in the C3(1)/SV40Tag transgenic mouse model of breast cancer [56]. On the other hand, Srivastava and Gupta [57] demonstrated that apigenin-7-*O*-glucoside, extracted from *M. chamomilla*, has a significantly higher anticancer activity compared to other glucoside derivatives tested. The results of another study indicated that apigenin possesses anticancer activity against the PC-3 lineage [53]. The flavone luteolin also exhibited potent anticancer activity [58]. This flavone suppresses a wide range of malignant tumors and cancer cell growth, including glioblastoma, breast, colon, pancreatic, prostate, oral, kidney, liver, ovarian, gastric, and skin cancers [59]. Luteolin can induce cancer cell apoptosis by phosphorylating JNK and inhibiting translocation of NF-κB as a transcription factor from the nucleus [58]. On the other hand, vanillin was reported as anticarcinogenic [60].

Based on the results of GC-MS analysis, the present study reveals that the methanolic extracts of *C. mixtus* contain terpenoids, fatty acids, esters, alcohols, alkanes, organic acids, phenols, and other compounds, with terpenoids compounds in abundance. Similarly, using GC-MS, methanolic flower extract of chamomile (*Matricaria aurea*) from Saudi Arabia showed the presence of several compounds belonging to aromatic compounds, phenols, alcohols, esters, ketones, and hydrocarbons. However, the detected molecules were different from the ones reported in our study [61]. In another work on aqueous extracts of two Moroccan chamomiles, *C. mixtus* and *M. chamomilla*, the phytochemical screening showed the presence of terpenoids, alkaloids, flavonoids, saponins, and tannins, while it showed the absence of anthraquinones [62].

Terpenoids are isoprene molecules that are important not only for plant growth and ecology but also provide a shield against insects. Terpenoids extracted from plants are used in the food, chemical, and pharmaceutical industries and also in the development of biofuel products [63]. Several studies have reported the antioxidant and anticarcinogenic activities of terpenoids [64]. For example, the diterpene phytol, dominant in extracts from the leaves and stems of our plant, has shown anticancer, antioxidant, anti-inflammatory, diuretic, antitumor, chemopreventive, and antimicrobial properties and has also been used in training vaccines [65]. Babu and Jayaraman [66] demonstrated that β-sitosterol had various biological actions, including anticancer effects. This compound was detected in the flowers of *C. mixus*. Additionally, among the various fatty acids detected in our extracts, palmitic acid was found in the leaves and flowers. This compound has also shown anticancer effects [67]. Consequently, we have found that *C. mixtus* contains many biomolecules known for their anticancer effects, explaining the promising results obtained against the selected cell lines.

In the future, these extracts can be further studied in other cancer cells, such as colon or lung cancer cells, or even in other cancer cell subtypes, such as more aggressive breast cancer cells (MDA-MB-231), to determine if their anticancer effect can be expanded to other types of cancer. Moreover, the exposure time of cancer cells to these extracts can be further explored to determine if some extracts may also be active at a lower concentration if the treatment period is increased. Mechanistic studies should also be performed to determine the accurate mechanism of action of these extracts in cancer cells, including docking simulation.

## 5. Conclusions

It can be concluded that *C. mixtus* is very rich in biochemical compounds that have various biological potentials, including anticancer activity. The data obtained in this study suggest that the majority of methanolic extracts of *C. mixtus* have antiproliferative activity against the two cell lines, human mammary carcinoma (MCF-7) and human prostate carcinoma (PC-3), in vitro using the MTT assay. The activity was dose-dependent, with the highest antiproliferative activity obtained by leaf extract against the two cell lines. Moreover, these extracts seem to be safe for normal cells. In conclusion, *C. mixtus* can be considered an anticancer herb since it contains various biochemical groups such as flavonoids, phenolic acids, terpenoids, and fatty acids that present interesting anticancer effects.

## Figures and Tables

**Figure 1 cancers-15-00152-f001:**
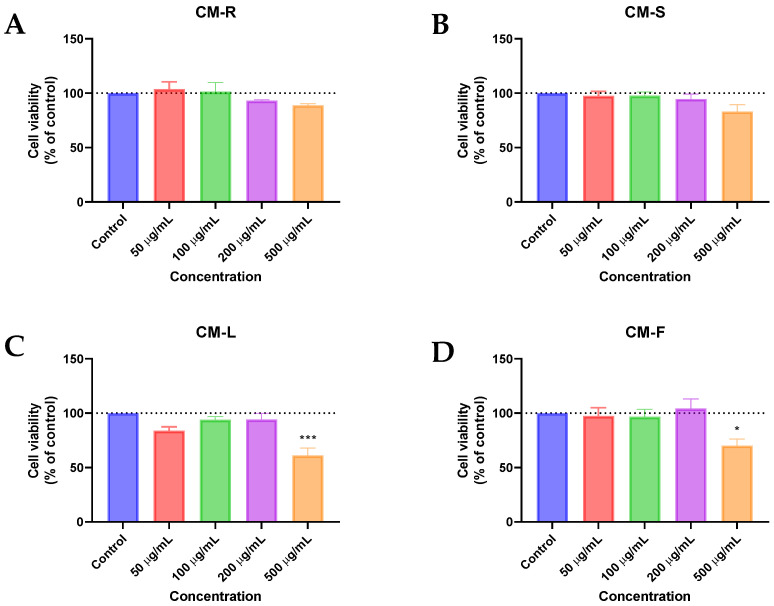
Biosafety evaluation of *Cladanthus mixtus* extracts on MRC-5 cells after 48 h of treatment. (**A**): root extract, (**B**): stem extract, (**C**): Leaf extract, (**D**): flower extract. Data are given as the mean ± SEM (n = 3), * statistically significant vs. control (vehicle) at *p* < 0.05; *** statistically significant vs. control (vehicle) at *p* < 0.001.

**Figure 2 cancers-15-00152-f002:**
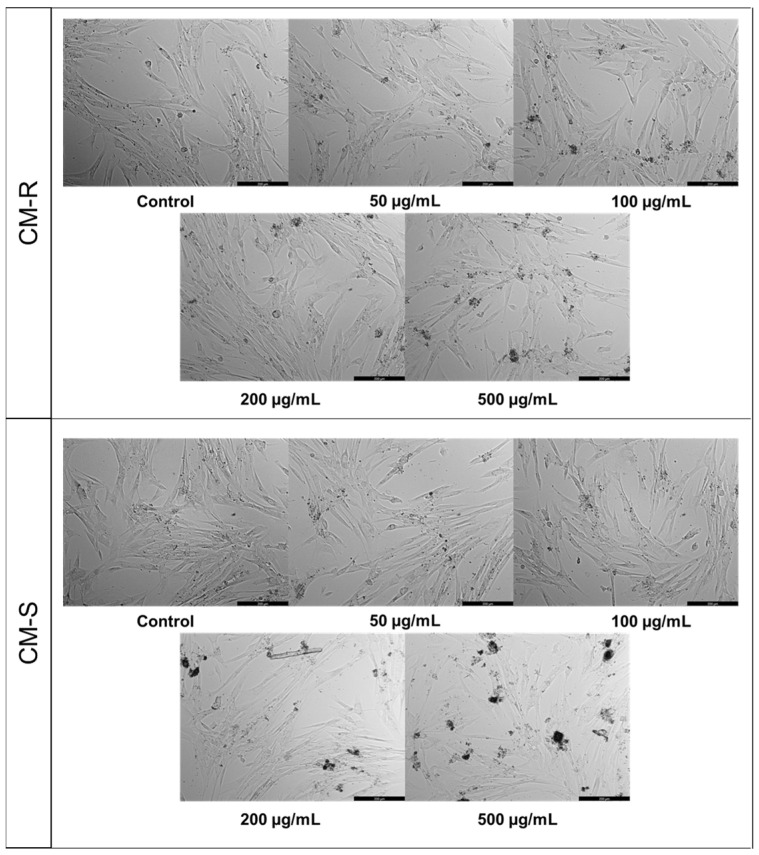
Morphological evaluation of MRC-5 cells after 48 h of treatment with *Cladanthus mixtus* extracts. Results are representative of three independent experiments. Scale bar: 200 µm.

**Figure 3 cancers-15-00152-f003:**
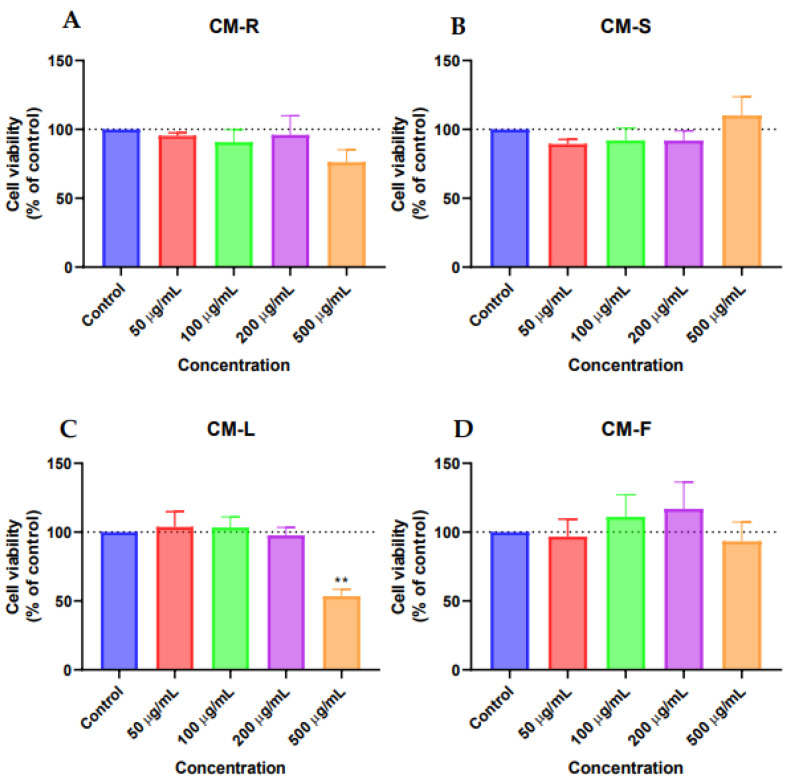
Cytotoxic evaluation of *Cladanthus mixtus* extracts on MCF-7 cells after 48 h of treatment. (**A**): root extract, (**B**): stem extract, (**C**): leaf extract, (**D**): flower extract. Data are given as the mean ± SEM (n = 3), ** statistically significant vs. control (vehicle) at *p* < 0.01.

**Figure 4 cancers-15-00152-f004:**
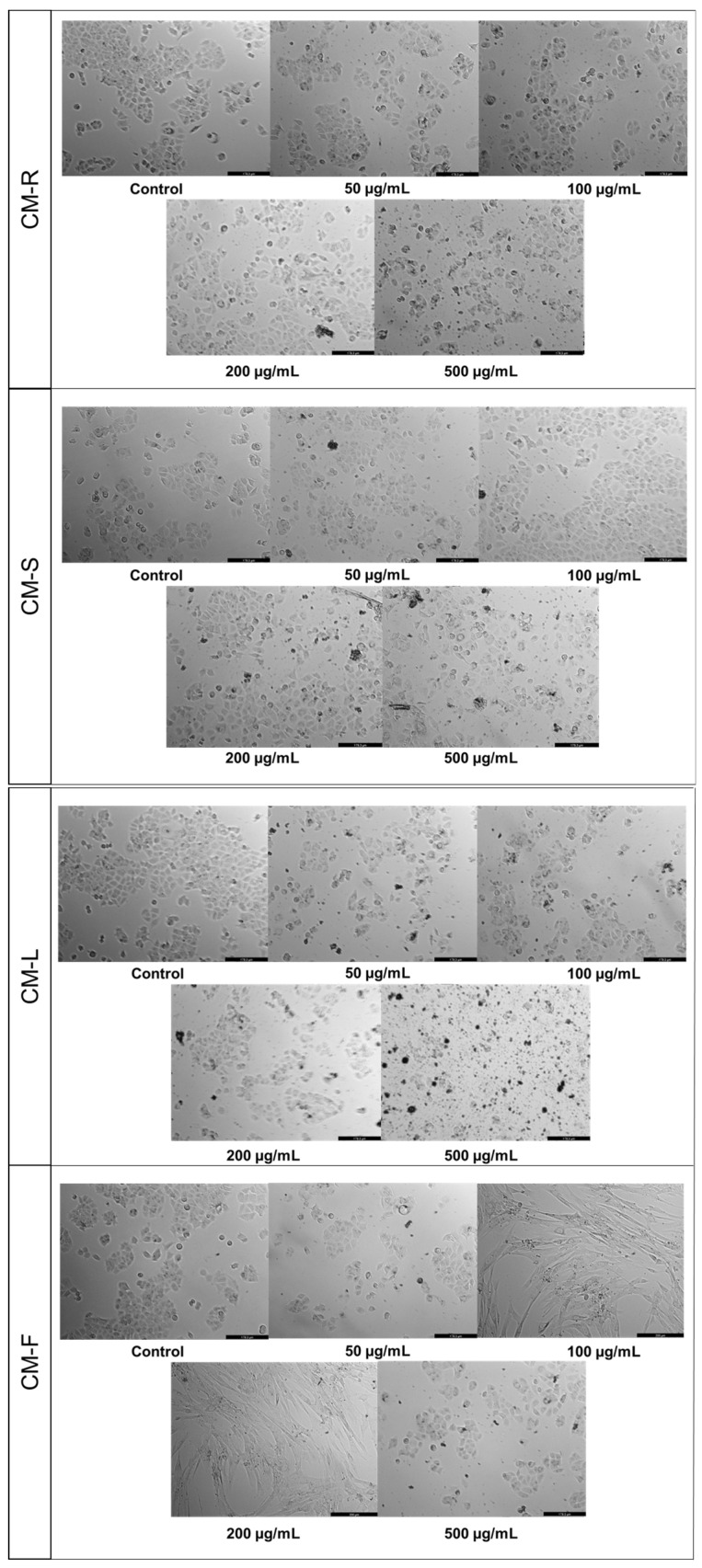
Morphological evaluation of MCF-7 cells after 48 h of treatment with *Cladanthus mixtus* extracts. Results are representative of three independent experiments. Scale bar: 200 µm.

**Figure 5 cancers-15-00152-f005:**
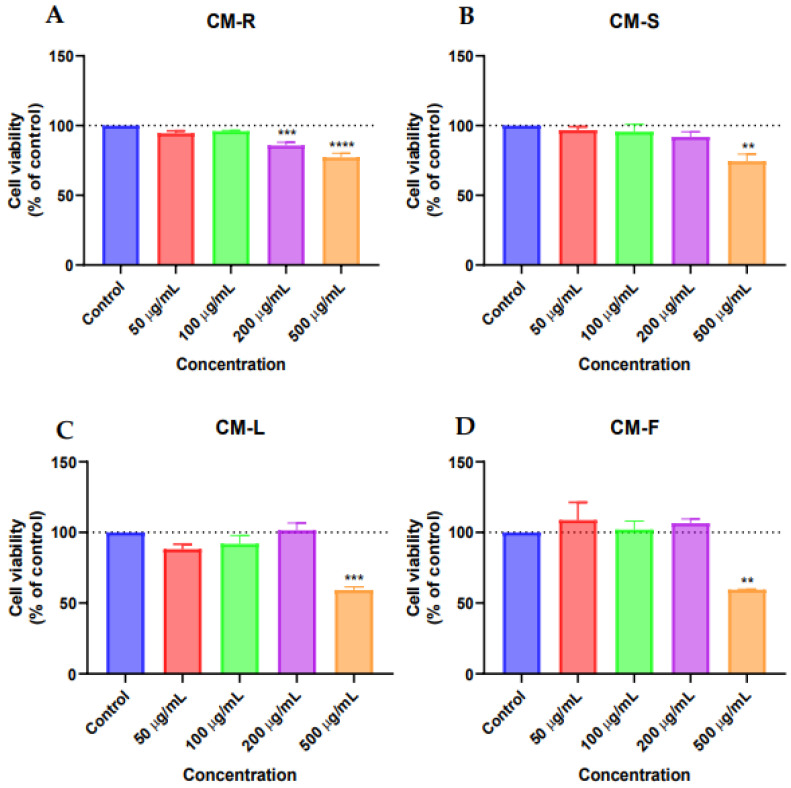
Cytotoxic evaluation of *Cladanthus mixtus* extracts on PC-3 cells after 48 h of treatment. (**A**): root extract, (**B**): stem extract, (**C**): leaf extract, (**D**): flower extract. Data are given as the mean ± SEM (n = 3), ** statistically significant vs. control (vehicle) at *p* < 0.01; *** statistically significant vs. control (vehicle) at *p* < 0.001; **** statistically significant vs. control (vehicle) at *p* < 0.0001.

**Figure 6 cancers-15-00152-f006:**
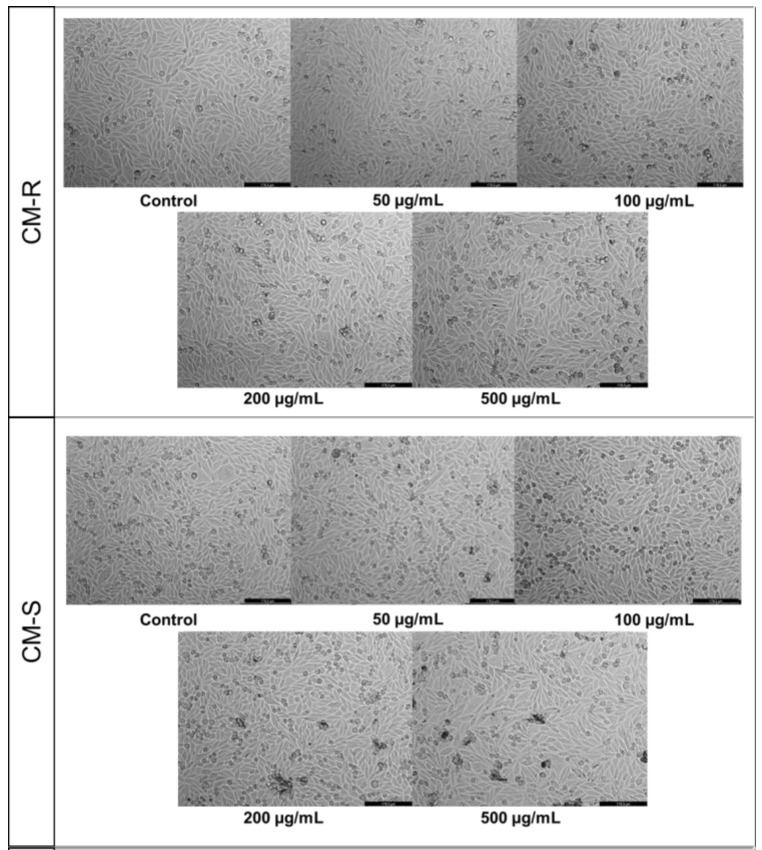
Morphological evaluation of PC-3 cells after 48 h of treatment with *Cladanthus mixtus* extracts. Results are representative of three independent experiments. Scale bar: 200 µm.

**Figure 7 cancers-15-00152-f007:**
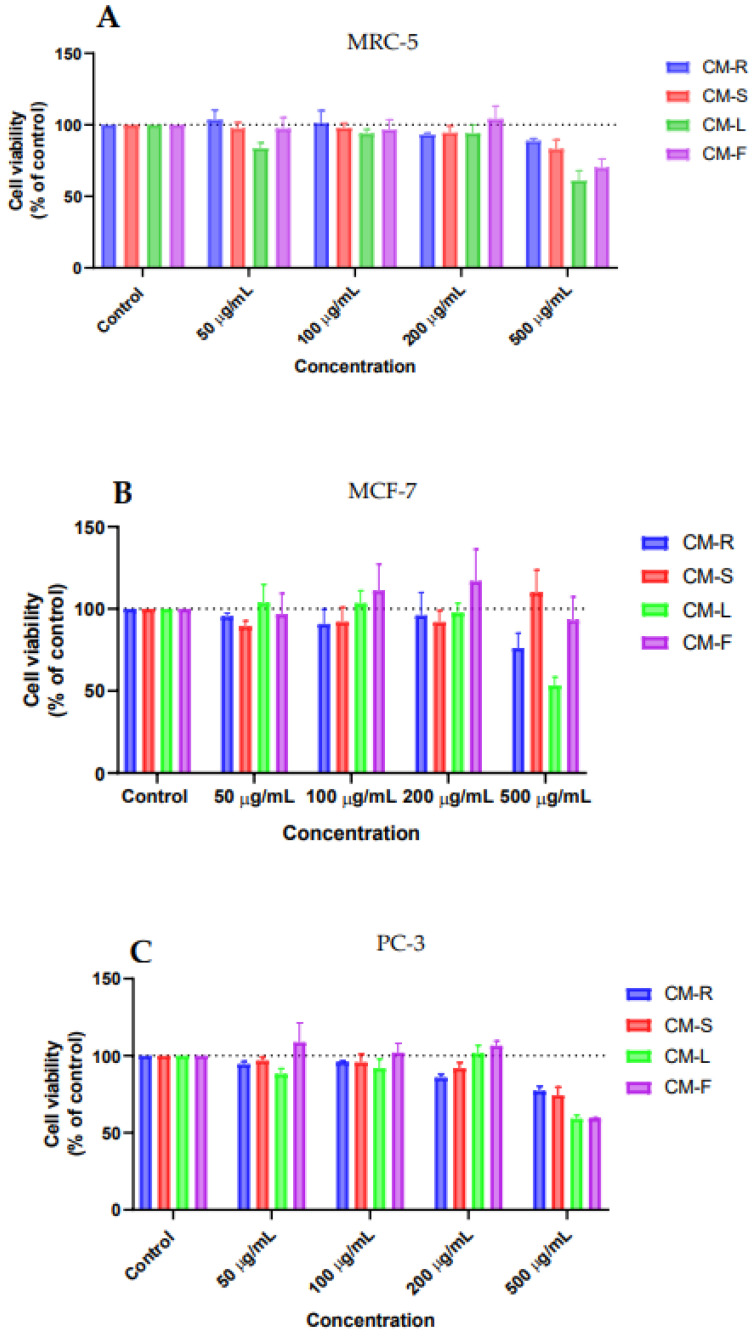
Comparative biosafety evaluation of the antitumor activity of *Cladanthus mixtus* extracts on cancer cells after 48 h of treatment. (**A**): MRC-5, (**B**): MCF-7, (**C**): PC-3.

**Table 1 cancers-15-00152-t001:** Quantification of phenolic compounds in extracts of *Cladanthus mixtus* obtained by HPLC-MS.

		Content (µg/g DW)			
	Compounds	Flowers (CM-F)	Leaves (CM-L)	Stems (CM-S)	Roots (CM-R)
**Phenolic acids**	Gallic acid	13.23 ± 0.12 ^q^	11.17 ± 0.26 ^o^	7.14 ± 0.07 ^q^	9.55 ± 0.19 ^q^
	Protocatechuic acid	216.11 ± 0.32 ^n^	41.06 ± 0.43 ^m^	124.29 ± 0.96 ^k^	96.68 ± 0.93 ^kl^
	Chlorogenic acid	1987.02 ± 0.69 ^a^	348.42 ± 0.60 ^e^	894.49 ± 0.02 ^q^	561.82 ± 0.94 ^b^
	Salicylic acid	318.11 ± 0.41 ^j^	133.35 ± 0.11 ^j^	122.05 ± 0.84 ^k^	110.20 ± 0.66 ^ij^
	*p*-Hydroxybenzoic acid	701.10 ± 0.04 ^f^	65.89 ± 0.35 ^l^	175.49 ± 0.02 ^i^	184.41 ± 0.07 ^g^
	Caffeic acid	206.71 ± 0.50 ^n^	126.80 ± 0.98 ^j^	123.02 ± 0.21 ^k^	86.29 ± 0.13 ^lm^
	Vanillic acid	--	77.51 ± 0.52 ^l^	192.89 ± 0.12 ^h^	130.13 ± 0.46 ^h^
	Syringic acid	345.23 ± 0.60 ^i^	146.81 ± 0.20 ^i^	135.31 ± 0.39 ^j^	58.94 ± 0.90 ^o^
	Rosmarinic acid	102.10 ± 0.72 ^p^	38.04 ± 0.23 ^m^	14.82 ± 0.23 ^pq^	11.06 ± 0.15 ^q^
	Ellagic acid	1666.22 ± 0.5 ^b^	1095.01 ± 0.43 ^a^	470.40 ± 0.56 ^d^	415.21 ± 0.53 ^c^
	*p*-Coumaric acid	407.13 ± 0.77 ^h^	191.44 ± 0.70 ^g^	92.76 ± 0.26 ^m^	83.05 ± 0.27 ^m^
	Methyl paraben	265.61 ± 0.35 ^l^	16.63 ± 0.15 n ^o^	18.73 ± 0.91 ^p^	34.71 ± 0.81 ^p^
	Ferulic acid	161.34 ± 0.16 ^o^	105.45 ± 0.46 ^k^	102.22 ± 0.76 ^l^	82.04 ± 0.75 ^m^
**Flavonoids**	Luteolin-7-*O*-glucoside	836.30 ± 0.38 ^e^	424.07 ± 0.08 ^c^	332.46 ± 0.59 ^e^	305.12 ± 0.88 ^d^
	Apigenin-7-*O*-glucoside	1074.03 ± 0.21 ^d^	958.59 ± 0.41 ^b^	819.60 ± 0.30 ^b^	603.02 ± 0.80 ^a^
	Luteolin	257.54 ± 0.71 ^m^	169.50 ± 0.33 ^h^	84.25 ± 0.49 ^m^	67.38 ± 0.88 ^no^
	Apigenin	119.14 ± 0.42 ^p^	37.14 ± 0.53 ^m^	29.83 ± 0.12 ^o^	54.01 ± 0.51 ^o^
	Quercetin	1292.01 ± 0.85 ^c^	23.38 ± 0.80 ^n^	307.33 ± 0.86 ^f^	73.63 ± 0.63 ^mn^
	Rutin	673.12 ± 0.99 ^g^	377.05 ± 0.55 ^d^	565.20 ± 0.58 ^c^	241.15 ± 0.53 ^f^
	Naringin	298.30 ± 0.02 ^jk^	220.61 ± 0.37 ^f^	141.63 ± 0.70 ^j^	105.61 ± 0.18 ^jk^
	Catechin	347.71 ± 0.22 ^i^	114.42 ± 0.74 ^k^	240.41 ± 0.12 ^g^	268.40 ± 0.28 ^e^
	Vanillin	141.94 ± 0.87 ^o^	138.04 ± 0.60 ^ij^	140.11 ± 0.79 ^j^	122.73 ± 0.71 ^hi^
	Kaempferol	290.60 ± 0.03 ^k^	69.57 ± 0.02 ^l^	72.98 ± 0.19 ^n^	39.90 ± 0.53 ^p^
	Isorhamnetin	237.53 ± 0.64 ^m^	110.90 ± 0.78 ^k^	108.72 ± 0.92 ^l^	129.65 ± 0.80 ^h^

Values are means ± standard deviation of three replicates. (--): Not detected. The letters a to q represent a significant difference between compounds in the same organ at *p* < 0.05.

**Table 2 cancers-15-00152-t002:** Biomolecule groups of *Cladanthus mixtus* extracts obtained by GC-MS.

	Area (%)			
Compound Groups	Flowers (CM-F)	Leaves (CM-L)	Stems (CM-S)	Roots (CM-R)
Terpenoids	10.41	46.20	11.32	10.88
Esters	13.88	13.21	30.11	21.91
Alcohols	--	6.74	5.05	24.49
Alkanes	15.14	2.63	18.13	3.14
Fatty acids	27.86	11.88	6.18	8.23
Organic acids	3.07	1.85	2.74	4.7
Benzene and its derivatives	2.85	7.62	5.73	8.15
Phenols	0.41	0.52	1.61	6
Ketones	--	1.25	4.47	1.67
Carbonyls	19.27	1.34	--	--
Amines	1.62	2.32	6.9	0.7
Pyrrolidines/pyrimidines	0.97	0.22	3.04	3.36
Steroids	--	2.32	--	--
Others	4.52	1.88	4.72	6.77
Total	100	99.98	100	100

(--): Not detected.

## Data Availability

The dataset generated and/or analyzed in this study are available upon reasonable request.

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
