# Peer review of "Phytochemical Compounds and Anticancer Activity of *Cladanthus mixtus* Extracts from Northern Morocco"

_cancers, 2022, doi:10.3390/cancers15010152_

Round 1
Reviewer 1 Report
Minor comments:
The manuscript requires extensive English rewriting (for example due to oversimplification and lack of formality). Terms including but not restricted to "a terrifying illness", "promising possibilities" and "pleiotropic effects on target events in many way" should be avoided.
In the introduction, the Authors wrote that one of their aims was to demonstrate that the tested extracts of C. mixtus caused tumor cell growth suppression and probably their apoptosis. In my opinion, this goal was not achieved, or was poorly formulated because none of the studies conducted allowed to determine the type of cell death.
Line 42 the word "human" is repeated twice "A human normal human lung fibroblast"
2.6.3 - Replace "cellular morphology" with "cell morphology in monolayer culture"
2.6.4 I propose to replace "biosafety profile" with "cytotoxicity" or "cytotoxic profile" - in my opinion, the name in its current form refers only to normal cells
2.7 In the section on statistics Authors refer only to the biological activity in cancer cells ("the mean ± SEM for anticancer activity"), what about normal cells? - incorrectly formulated
3.3 line 338-9 In the description of the conclusions it is incorrect to state that the tested extracts are safe for normal cells at a concentration below 500 µg/mL. CM-L and CM-F extracts at a concentration of 500 µg/mL caused a statistically significant decrease in cell viability, while the previous concentration was 200 µg/mL and did not cause a significant decrease in cell viability. The authors did not test concentrations of, for example, 300/400 µg/mL, therefore they cannot be sure of the reaction of cells to such concentrations of these extracts. Therefore, based on the presented results, it can be concluded that concentrations below 200 µg/mL are certainly safe.
Figures 3 and 5 should not be labeled "biosafety" as they refer to cancer cells
Fig 3 B and D - how the authors explain the increase in cell viability after treatment with CM-S at a concentration of 500 µg/mL and CM-F - 100 and 200 µg/mL?
Fig 7. The name of the MRC-5 line above the graph is missing
Major complaint: The authors drew incompletely correct conclusions from the obtained in vitro studies (line 392-408). The extracts that the Authors indicate as the most promising (CM-L and CM-F) are also the most cytotoxic to normal cells. Both extracts work most effectively in cancer cells, in the same concentrations, in which a significant decrease in the survival of MRC-5 cells was obtained. Moreover, cell survival of the MCF-7 line was better than that of the MRC-5 line. Only cells of the PC-3 line showed greater sensitivity to the tested extracts. In my opinion, special attention should be paid to the CM-R extract, which was the only one that generated a decrease in cancer cell viability already at a concentration of 200 µg/mL, while being safe for normal MRC-5 cells.
Author Response
Review 1
The manuscript requires extensive English rewriting (for example due to oversimplification and lack of formality). Terms including but not restricted to "a terrifying illness", "promising possibilities" and "pleiotropic effects on target events in many way" should be avoided.
Thank for your suggestion. The text has been thoroughly revised and some expressions improved.
In the introduction, the Authors wrote that one of their aims was to demonstrate that the tested extracts of C. mixtus caused tumor cell growth suppression and probably their apoptosis. In my opinion, this goal was not achieved, or was poorly formulated because none of the studies conducted allowed to determine the type of cell death.
Thank you for your note. You are right, that goal has been removed from the text.
Line 42 the word "human" is repeated twice "A human normal human lung fibroblast"
The mistake has been corrected.
2.6.3 - Replace "cellular morphology" with "cell morphology in monolayer culture"
Thank you for your suggestion.
2.6.4 I propose to replace "biosafety profile" with "cytotoxicity" or "cytotoxic profile" - in my opinion, the name in its current form refers only to normal cells
Thank you for your suggestion.
2.7 In the section on statistics Authors refer only to the biological activity in cancer cells ("the mean ± SEM for anticancer activity"), what about normal cells? - incorrectly formulated
The mistake has been corrected. Now appears as: The results of three independent experiments are represented as the mean ± SEM for antiproliferation activity and ± SD for chemical analysis in the three cell types assayed.
3.3 line 338-9 In the description of the conclusions it is incorrect to state that the tested extracts are safe for normal cells at a concentration below 500 µg/mL. CM-L and CM-F extracts at a concentration of 500 µg/mL caused a statistically significant decrease in cell viability, while the previous concentration was 200 µg/mL and did not cause a significant decrease in cell viability. The authors did not test concentrations of, for example, 300/400 µg/mL, therefore they cannot be sure of the reaction of cells to such concentrations of these extracts. Therefore, based on the presented results, it can be concluded that concentrations below 200 µg/mL are certainly safe.
Thank you for your appointment. We correct this aspect.
Figures 3 and 5 should not be labeled "biosafety" as they refer to cancer cells
The suggestion has been incorporated as in 2.6.4.
Fig 3 B and D - how the authors explain the increase in cell viability after treatment with CM-S at a concentration of 500 µg/mL and CM-F - 100 and 200 µg/mL?
Obviously, these results are surprising. However, we think that these small increases in cell viability over the mean data are not statistically significant, although they may indicate that some of its constituents could present pro-activity.
Fig 7. The name of the MRC-5 line above the graph is missing
The mistake has been corrected.
Major complaint: The authors drew incompletely correct conclusions from the obtained in vitro studies (line 392-408). The extracts that the Authors indicate as the most promising (CM-L and CM-F) are also the most cytotoxic to normal cells. Both extracts work most effectively in cancer cells, in the same concentrations, in which a significant decrease in the survival of MRC-5 cells was obtained. Moreover, cell survival of the MCF-7 line was better than that of the MRC-5 line. Only cells of the PC-3 line showed greater sensitivity to the tested extracts. In my opinion, special attention should be paid to the CM-R extract, which was the only one that generated a decrease in cancer cell viability already at a concentration of 200 µg/mL, while being safe for normal MRC-5 cells.
Thank you for your suggestion. We added these considerations in Discussion section of the new version.
Reviewer 2 Report
1. The simple summary is not oriented to the point. It must be well described explaining your final result or your output.
2. Line 33: "four extracts" is a confused word as it means four different extracts of different plants. So, I prefer to change it.
3. Line 47: your result is not clear "The majority of extracts". Please, mention which part gives the best result and its dose.
4. Introduction: I suggest a brief sentence about breast cancer and also prostate cancer that were handled in your study.
5. Line 89: What are these traditional uses or ailments? & What is the traditional part used?
6. Line 122: The authors should thank Prof. Lamarti in acknowledgment mentioning his last name and his affiliation or speciality.
7. Line 129 and 222: Cladanthus mixtus should be in italic font.
8. Line 136: I don't understand the importance of liquid-liquid extraction step, and which fraction was subjected to HPLC-MS analysis?
9. Line 144: "the phenolic part". How did you separate or prepare this fraction in particular?
10. Line 223: "HPLC-MS analysis". You must provide the chromatograms of HPLC-MS analysis of extracts of different parts
11. How did you identify the metabolites mentioned in table 1? If you did it using HPLC-MS, you must provide mass fragments and calculate mass error. If you did it using HPLC only, you must provide the chromatograms of different standards matched with peaks?
12. In table 1: the letters should be in superscript.
13. There is not a big difference in chemical composition of different parts. Why did the leaf extract display the most significant antitumor effect against cancer cell lines? You have to explain your results and correlate between them.
14. It is important to make a control group to detect the false effect of a vehicle (DMSO) but also you must compare your results with a standard /drug to determine the positive effect of your plant.
15. The first paragraph in discussion part is too long. It could be concise.
16. You mention that "this is the first study on your plant in Morocco. Please, compare your findings with other previous studies of the same genus in different regions in your discussion part.
17. Line 417: The hydroalcoholic solution extracted all metabolites including phenolics, glycosides, ceramides, alkaloids, etc. So, why did you mention phenolics in particular?
18. I recommend making a docking simulation of detected metabolites to your ligands to illustrate or prove your results.
Author Response
see file

Reviewer 3 Report
The article is very well written. The text is simple and easy to read. The discussion and conclusions are consistent with the results and arguments presented. Please use your own words on the methodology and discussion to reduce the percentage of plagiarism.
Author Response
Review 3
The article is very well written. The text is simple and easy to read. The discussion and conclusions are consistent with the results and arguments presented. Please use your own words on the methodology and discussion to reduce the percentage of plagiarism.
Thank you for your evaluation. We have reformulated the text to avoid the plagiarism.
Reviewer 4 Report
This original research article entitled “Phytochemical compounds and anticancer activity of Cladanthus mixtus extracts from northern Morocco” submitted by El Mihyaoui and all., presents the phytochemical composition of Cladanthus mixtus, a plant used in traditional Moroccan medicine and its potential as source of anticancer molecules. Authors prepared methanolic extracts from different parts of the plant (roots, stems, leaves, flowers) and analyzed their chemical composition by HPLC and GC-MS. Next, they evaluated the potential of the different extracts by MTT assay against breast and prostate human cancer cells (respectively MCF-7 and PC-3 cell lines), and human normal lung fibroblast (MRC-5). Results show that extracts prepared from root and leaves exert cytotoxic activity and may therefore be considered for discovery of anticancer agents.
The manuscript is clear and well written. However, in its present form, the manuscript needs major modifications before acceptance for publication, especially because it DOES NOT followed instructions to authors instructed by Cancer journal (https://www.mdpi.com/journal/cancers/instructions) and as mentioned “Editors reserve the rights to reject any submission that does not meet these requirements.”. The quality of the microscopy figures must also be improved.
Modifications:
Line 58 - 64: Please update your manuscript with more recent statistics (Cancer Statistics 2022, https://doi.org/10.3322/caac.21708).
Line 119:
As mentioned in the instructions to authors (https://www.mdpi.com/journal/cancers/instructions), for research involving plants, a voucher specimen must be deposited in an accessible herbarium/museum and information must be provided (GPS coordinates, population,...).
Line 129:
Please mention if extraction was performed under agitation or not.
Line 141:
At which concentration the dry extracts were resuspended for HPLC-MS analysis and which solvent was used?
Line 174:
How did you identify the compounds? Did you perform comparative analysis using chemical standards or did you refer to a house / commercial library. Please include the information.
At which concentration did you resuspend the dry extracts in chloroform for analysis?
Line 202: In which conditions did you perform morphological analysis? Did you seed cells in 6-well plates or cell culture chambers. At which time point (48h?) did you observe the cells.
Figure 1 to 7:
Please add that MTT assay was performed 48h after treatment.
For microscopy figure (figure 2, 4 and 6), the scale bar is absent/partially masked.
The quality of the pictures must be improved. At 100%, on a laptop monitor, pictures are blurred. Please submit better quality and resolution images.
Line 332:
The expression “little decrease” is not scientific, please mention the percentage difference observed.
Line 336:
You mention “...a suppression of proliferation and fewer cell number...” but as I mention above, the quality of the microscopy pictures is not good enough to see and validate your observation for a reader. I would also recommend to include cell quantification (blue trypan, imageJ counting,...) with quantification/plot to confirm your observations.
Line 353:
You mention “...cells were rounder and in less number...” but as I mention above, the quality of the microscopy pictures is not good enough to see and validate your observation for a reader. Quantification / cell density should also be included.
Line 410 - Figure 7
Title for MRC-5 cells plot is missing.
Comments:
You performed all your cytototxic analysis at 48 hours. Did you extend your analysis up to 72h post-treatment to see if some extracts may also be active at a lower concentration?
In your discussion, you mention many times that some phytochemicals are able to induce apoptosis in different cancer cells lines. Why did not you analyze if your extracts may also have some apoptotic potential using fluorescence techniques (DAPI staining,...) or commercial kits?
Author Response
Review 4
The manuscript is clear and well written. However, in its present form, the manuscript needs major modifications before acceptance for publication, especially because it DOES NOT followed instructions to authors instructed by Cancer journal (https://www.mdpi.com/journal/cancers/instructions) and as mentioned “Editors reserve the rights to reject any submission that does not meet these requirements.”. The quality of the microscopy figures must also be improved.
We thank the reviewer for pointing this out. As suggested by the reviewer, we have replaced these figures to improve their quality and the present revised form of our manuscript fits with the Cancers journal instructions.
Modifications:
Line 58 - 64: Please update your manuscript with more recent statistics (Cancer Statistics 2022, https://doi.org/10.3322/caac.21708).
Thank you for your suggestion. We have read the article that you proposed https://doi.org/10.3322/caac.21708) and we have found that it only reports the statistics of the United States. We have reported the worldwide statistics, we have checked again and there is just the 2020 statistics.
Line 119:
As mentioned in the instructions to authors (https://www.mdpi.com/journal/cancers/instructions), for research involving plants, a voucher specimen must be deposited in an accessible herbarium/museum and information must be provided (GPS coordinates, population,...).
Thank you for your comment.
We have incorporated, as requested by editor, an Ethical Statements section:
Cladanthus mixtus (L.) Chevall. plants were collected at full maturity from Beni Hassane (Tanger-Tetouan-Al Hoceima region, northern Morocco; GPS: N 35° 21’ 20.865” W 5° 22’ 12.677”), identified and deposited in the herbarium of the Faculty of Sciences, Abdelmalek Essaadi University, by Prof. A. Lamarti in Tetouan (Morocco). Botanical characteristics can be consulted at taxon page of World Floral On-line (WFO, 2022): Cladanthus mixtus (L.) Chevall. Published on the Internet; http://www.worldfloraonline.org/taxon/wfo-0000122557. Accessed on: 23 Nov 2022.
Human cell lines were provided by the American Type Culture Collection company (ATCC; Manassas, VA, USA), including MRC-5 human normal lung fibroblast, MCF-7 human mammary carcinoma, and PC-3 human prostate carcinoma cell lines.
Line 129:
Please mention if extraction was performed under agitation or not.
Thank you for your remark. The extraction was not performed under agitation. For this reason, we did not mention it in the text.
Line 141:
At which concentration the dry extracts were resuspended for HPLC-MS analysis and which solvent was used?
Thank you for your question. The concentration was 10 mg of dry extract per 1 mL of extraction solvent methanol:water (80:20) (Lines 156-157)
Line 174:
How did you identify the compounds? Did you perform comparative analysis using chemical standards or did you refer to a house / commercial library. Please include the information.
At which concentration did you resuspend the dry extracts in chloroform for analysis?
Thank you for your pertinent question. As described in subsection “2.3.2. Analysis by HPLC-MS” the identification of the compounds was done by using UV-Vis spectra, mass spectra and retention times of each compound analyzed and comparing them to the standards ones which were analyzed using the same conditions and equipment. The concentration was 10 mg of dry extract per 1 mL of chloroform (Line 184).
Line 202: In which conditions did you perform morphological analysis? Did you seed cells in 6-well plates or cell culture chambers. At which time point (48h?) did you observe the cells.
Thank you for your pertinent question. Morphological analysis was performed after a 48 h treatment with each extract, before performing the MTT assay, using 96-well plates. In this way, morphological analysis and MTT assays were performed using the same plates, in the same day. We have included this information in the main text.
Figure 1 to 7:
Please add that MTT assay was performed 48h after treatment.
For microscopy figure (figure 2, 4 and 6), the scale bar is absent/partially masked.
The quality of the pictures must be improved. At 100%, on a laptop monitor, pictures are blurred. Please submit better quality and resolution images.
Thank you for pointing this out. We have added that MTT and microscopic observations were done after 48h in all figure’s titles (from 1 to 7). Concerning the figure’s quality, we agree with the reviewer and have replaced these figures.
Line 332:
The expression “little decrease” is not scientific, please mention the percentage difference observed.
Thank you for your suggestion. We agree and have changed this sentence (Line 336).
Line 336:
You mention “...a suppression of proliferation and fewer cell number...” but as I mention above, the quality of the microscopy pictures is not good enough to see and validate your observation for a reader. I would also recommend to include cell quantification (blue trypan, imageJ counting,...) with quantification/plot to confirm your observations.
Thank you for pointing this out. As mentioned above, we agree with the reviewer and have replaced these figures. We have not performed cell counting using tripan blue because we performed the morphological analysis using the same plates designated for the MTT assay. ImageJ counting was not possible due to the atypical morphology of MRC-5 cells.
Line 353:
You mention “...cells were rounder and in less number...” but as I mention above, the quality of the microscopy pictures is not good enough to see and validate your observation for a reader. Quantification / cell density should also be included.
Thank you for your suggestion. We agree with the reviewer and have replaced these figures. We have not performed cell counting using tripan blue because we performed the morphological analysis using the same plates designated for the MTT assay. ImageJ counting was not possible due to the atypical morphology of MRC-5 cells.
Line 410 - Figure 7
Title for MRC-5 cells plot is missing.
Thank you for pointing this out. We have added the plot.
Comments:
You performed all your cytotoxic analysis at 48 hours. Did you extend your analysis up to 72h post-treatment to see if some extracts may also be active at a lower concentration?
Thank you for your suggestion. To our knowledge, this is the first study ever performed using Cladanthus mixtus from Northern Morocco on human tumor cells. Therefore, we believe that for a preliminary study it would be more appropriate to select just one intermediate time point to perform the in vitro experimentation, as we included one non-tumoral cell line and two cancer cell lines. We believe that it remains a convenient starting point for discovery and proof-of-concept studies like this one. We have included this information in the discussion:
“In the future, these extracts can be further studied in other cancer cells, such as colon or lung cancer cells or even in other cancer cell subtypes, such as more aggressive breast cancer cells (MDA-MB-231), to determine if their anticancer effect can be expanded to other types of cancer. Moreover, the exposure time of cancer cells to these extracts can be further explored, to determine if some extracts may also be active at a lower concentration if the treatment period is increased. Mechanistic studies should also be performed to determine the accurate mechanism of action of these extracts in cancer cells.”
In your discussion, you mention many times that some phytochemicals are able to induce apoptosis in different cancer cells lines. Why did not you analyze if your extracts may also have some apoptotic potential using fluorescence techniques (DAPI staining,...) or commercial kits?
Thank you for pointing this out. We agree that apoptotic studies would contribute for the enrichment of our manuscript. Nevertheless, as we mentioned above, this was a preliminary study and therefore we considered the MTT assays and the morphological analysis as the more convenient approaches for the evaluation of the anticancer effects of these extracts. In the future, we intend to perform other studies that include apoptotic assays as well as further mechanistic studies.
Round 2
Reviewer 2 Report
Thank you for your effort, but some comments are still unclear.
1. Liquid-liquid extraction is not considered as a purification step of phenolic. It is a fractionation step to prepare fractions of different polarities. Which organic layer was injected? It should be clear. I want to ask about this point "Why didn't you inject the methanolic extract to HPLC as a routine work in HPLC analysis?" Also, if you want to isolate a phenolic fraction, you can simply do it through acid-base extraction method?
2. This is the first study on this plant in your region. Please, you must compare your results of HPLC and GC-MS analysis with other chamomiles grown in different regional area (NOT in Morocco) in your discussion.
3. Docking simulation is not out of scope of this paper as it can explain the possible linkage between your identified phenolics or terpenoids, and the possible ligands, tumor marker and genes. If it is possible, it will proof your idea.
Author Response
Reviewer 2
Comments and Suggestions for Authors
Thank you for your effort, but some comments are still unclear.
- Liquid-liquid extraction is not considered as a purification step of phenolic. It is a fractionation step to prepare fractions of different polarities. Which organic layer was injected? It should be clear. I want to ask about this point "Why didn't you inject the methanolic extract to HPLC as a routine work in HPLC analysis?" Also, if you want to isolate a phenolic fraction, you can simply do it through acid-base extraction method?
We appreciate your suggestion and apologize for the unclear description. Methanol with water is often used to extract the phenolic compounds from plant materials in a crude form and liquid-liquid extraction was used to ensure the removal of unwanted interferences such as fats, carbohydrates, proteins, pigments, terpenes, and other non-phenolic compounds. As described in the subsubsection “2.3.1. Extraction of phenolic compounds from Cladanthus mixtus”, the organic phase (combined diethyl ether and ethyl acetate extracts) was evaporated until getting dried extracts and then dissolved in methanol:water (80:20) at a requested concentration (10 mg/ml) and filtered through a 0.22 μm disposable LC filter disk for HPLC analysis.
The information needed was added to the “2.3.1. Extraction of phenolic compounds from Cladanthus mixtus”, lines 159-160.
- This is the first study on this plant in your region. Please, you must compare your results of HPLC and GC-MS analysis with other chamomiles grown in different regional area (NOT in Morocco) in your discussion.
Thank you for your comment. We have tried to discuss the chemical composition with works on other chamomiles as requested. We have added reference 32 in lines 886-889 to discuss more HPLC-MS analysis, and references 61 and 62 in lines 1157-1164 for GC-MS analysis.
- Piri, E.; Sourestani, M.M.; Khaleghi, E.; Mottaghipisheh, J.; Zomborszki, Z.P.; Hohmann, J.; Csupor, D. Chemo-diversity and antiradical potential of twelve Matricaria chamomilla L. populations from Iran: Proof of ecological effects. Molecules. 2019, 24,1315. https://doi.org/10.3390/molecules24071315
- Rizwana, H.; Alwhibi, M.S.; Soliman, D.A. Research article antimicrobial activity and chemical composition of flowers of Matricaria aurea a native herb of Saudi Arabia. Int. J. Pharmacol. 2016, 12, 576-586.
DOI: 10.3923/ijp.2016.576.586 - Hajjaj, G.; Bahlouli, A.; Sayah, K.; Tajani, M.; Cherrah, Y.; Zellou, A. Phytochemical screening and in-vivo antipyretic activity of the aqueous extracts of three Moroccan medicinal Plants. Pharm. Biol. Eval. 2017, 4(4), 88-92. DOI: http://dx.doi.org/10.26510/2394-0859.pbe.2017.30
- Docking simulation is not out of scope of this paper as it can explain the possible linkage between your identified phenolics or terpenoids, and the possible ligands, tumor marker and genes. If it is possible, it will proof your idea.
We thank the reviewer suggestion, and we will do docking simulations in the research in progress that will be the subject of a next work.
Reviewer 4 Report
After revision, all my comments and modifications were addressed by the authors.
I therefore recommend acceptance of the manuscript for publication after minor revisions and text editing.
Editing:
Line 30: please check “its”.
Line 103: please confirm “assalicylic acid”.
Modifications
Line 127:
Please mention that extraction was perform without agitation / in static condition.
As mentioned in the instructions to authors, material and methods section should provides enough details to allow an external researcher to repeat the methodology exactly without requiring further information.
Author Response
Reviewer 4
Comments and Suggestions for Authors
After revision, all my comments and modifications were addressed by the authors.
I therefore recommend acceptance of the manuscript for publication after minor revisions and text editing.
Editing:
Line 30: please check “its”.
Line 103: please confirm “assalicylic acid”.
Thank you for your comments:
We corrected “its” with “their”.
We corrected the word “assalicylic acid” with “as salicylic acid”.
Modifications
Line 127:
Please mention that extraction was perform without agitation / in static condition.
As mentioned in the instructions to authors, material and methods section should provides enough details to allow an external researcher to repeat the methodology exactly without requiring further information.
Thank you for your comment. We have added “in static condition” in line 134.